# The Impact of Silver Nanoparticles Functionalized with Spirulina Protein Extract on Rats

**DOI:** 10.3390/ph17091247

**Published:** 2024-09-22

**Authors:** Ludmila Rudi, Inga Zinicovscaia, Liliana Cepoi, Tatiana Chiriac, Dmitrii Grozdov, Alexandra Kravtsova

**Affiliations:** 1Institute of Microbiology and Biotechnology, Technical University of Moldova, MD 2028 Chisinau, Moldova; liliana.cepoi@imb.utm.md (L.C.); tatiana.chiriac@imb.utm.md (T.C.); 2Joint Institute for Nuclear Research, Joliot-Curie 6, 141980 Dubna, Russia; zinikovskaia@mail.ru (I.Z.); dsgrozdov@rambler.ru (D.G.); alexkravtsova@yandex.ru (A.K.); 3Horia Hulubei National Institute for RD in Physics and Nuclear Engineering, 30, Str. Reactorului, 077125 Magurele, Romania

**Keywords:** silver nanoparticles, biofunctionalization, Spirulina protein extract, rats, organ accumulation, hematological indicators, biochemical indicators

## Abstract

**Background/Objectives**: This study investigates the biocompatibility and physiological impacts of silver nanoparticles (AgNPs) functionalized with Spirulina protein extract (SPE) on laboratory rats. The objective was to assess and compare the systemic distribution, organ accumulation, and changes in hematological and biochemical parameters between biofunctionalized and non-functionalized silver nanoparticles. **Methods:** AgNPs were functionalized with SPE. Adult Wistar rats were administered these nanoparticles to assess their distribution across various organs using ICP-MS analysis. Hematological and biochemical markers were measured to evaluate systemic effects. **Results**: Functionalized silver nanoparticles demonstrated preferential accumulation in the brain, liver, and testicles, with significant clearance observed post-administration. The persistence of AgNPs SPE in reproductive organs was established. Hematological analysis revealed moderate changes, suggesting mild immune activation. Biochemical tests indicated transient increases in liver enzymes, signaling reversible hepatic stress. **Conclusions**: The biofunctionalization of AgNPs with Spirulina protein extract modifies the nanoparticles’ systemic behavior and organ distribution, enhancing their biocompatibility while inducing minimal physiological stress. These findings support the potential of Spirulina-based coatings to mitigate the toxicity and enhance the therapeutic efficacy of nanomedical agents.

## 1. Introduction

Silver nanoparticles have become an increasingly widespread area of research and application due to their essential capabilities in medicine, industry, and consumer products. Due to their antimicrobial efficacy, AgNPs are widely used in various products such as toothbrushes and toothpaste, food packaging, and others [1,2]. Additionally, AgNPs have shown potential as vaccine adjuvants, enhancing their effectiveness [3].

AgNPs, with their antibacterial, anti-inflammatory, and potential anticancer activities, are integrated into numerous medical fields [4]. Their role in combating multidrug-resistant bacteria is particularly noteworthy [5]. In wound treatment, AgNPs are a valuable tool used in dressings and gels to prevent infections and accelerate healing [6].

Due to their unique contrast properties, AgNPs are being explored in oncology as therapeutic agents for destroying cancer cells through cytotoxic mechanisms and medical imaging [7,8]. Additionally, in medical diagnostics, AgNPs are used in biosensors and diagnostic platforms to rapidly and precisely detect pathogens or specific biomarkers [7].

The therapeutic properties of AgNPs are influenced by their physicochemical characteristics, such as size, shape, size distribution, dose, origin, mode of administration, and specific surface area [9]. These characteristics determine their biological activity, stability in different environments, and ability to interact with molecules and cells. The functionalization of nanoparticles plays a crucial role in optimizing these properties, influencing their reactivity, stability, aggregation, bioavailability, and toxicity. For example, nanoparticles functionalized with thiobarbituric acid exhibit improved dispersibility and increased stability in aqueous solutions, while those functionalized with 11-mercaptoundecanoic acid and chitosan modify protein interactions and have slightly reduced toxicity [10,11].

By adjusting the size and coating of nanoparticles, it is possible to develop AgNPs that are optimized for specific applications [12]. Controlled surface functionalization of nanoparticles with various biomolecules, such as peptides or proteins, can prevent aggregation and enhance antimicrobial effects [13,14]. An effective method of biofunctionalizing silver nanoparticles is their synthesis using biological materials, where the reducing substrate largely determines the final properties of the AgNPs [15]. Different biological substrates can serve as matrices for synthesizing AgNPs with valuable properties. For example, nanoparticles synthesized using extracts from *Cuminum cyminum* L. seeds have shown reduced toxicity and anticancer properties [16]. Silver nanoparticles synthesized using extracts from the cyanobacterium *Spirulina platensis* have demonstrated antibacterial effects against *S. aureus* and *E. coli* [17]. Similarly, nanoparticles synthesized with aloe vera gel extract have exhibited antifungal effects and 100% inhibition of mice’s B16F10 melanoma cell line [18]. Silver nanoparticles obtained using flavonoid extracts from *Perilla frutescens* have significantly inhibited Gram-positive and Gram-negative bacterial pathogens and anticancer effects in colon cancer [19]. Nanoparticles synthesized using *Solanum nigrum* and *Pouteria sapota* extracts have shown potential for reducing blood sugar levels in diabetic animals [20]. The variability in the antibacterial effect of AgNPs is attributed to the biological components on the surface of the nanoparticles, which can influence their antibacterial mechanisms [5].

Conjugating AgNPs with bioactive molecules such as peptides, antibodies, or drugs is another crucial aspect of functionalization, essential for developing targeted delivery systems or enhancing therapeutic efficacy [21,22]. For example, conjugating AgNPs with Amoxicillin offers a promising strategy for combating antibiotic-resistant bacterial strains [23]. Silver nanoparticles conjugated with C-phycocyanin have shown effectiveness in healing dermal wounds, reducing erythrocyte toxicity, and benefiting from antibacterial properties [24].

*Spirulina platensis*, a cyanobacterium known for its antioxidant, anti-inflammatory, and immunomodulatory properties, is used in multiple biomedical applications [25,26]. Due to its rich composition of proteins, polysaccharides, and pigments, Spirulina reduces support for functionalizing and conjugating nanoparticles [27,28]. The interaction with nanoparticles allows Spirulina’s biological components to provide stability and biocompatibility while simultaneously reducing the potential toxicity of the nanoparticles [17].

This study investigated the accumulation characteristics and the changes in hematological and biochemical parameters in laboratory rats resulting from administering citrate-stabilized silver nanoparticles biofunctionalized with Spirulina protein extract, compared to non-functionalized citrate-stabilized nanoparticles. 

## 2. Results

### 2.1. Composition of Spirulina Protein Extract (SPE)

Table 1 presents the amino acid composition of the Spirulina protein extract (SPE). The extract contains a variety of essential amino acids (histidine, isoleucine, leucine, lysine, methionine, phenylalanine, threonine, tryptophan, and valine) and non-essential amino acids (alanine, arginine, aspartic acid, cysteine, glutamic acid, glycine, proline, serine, and tyrosine). The total extracted protein content was 52%. Non-essential amino acids constitute 68.68% (35.71 mg/100 mg extract), while essential amino acids comprise 31.32% (16.28 mg/100 mg extract).

Glutamic acid is the most abundant non-essential amino acid, at 9.04 mg/100 mg extract (17.39% of the total amino acids). The extract also contains significant amounts of arginine (5.58 mg/100 mg extract or 10.77%) and aspartic acid (5.02 mg/100 mg extract or 9.66%). Among the essential amino acids, leucine is in the highest concentration (5.12 mg/100 mg extract or 9.85%). Lysine (3.16 mg/100 mg extract or 6.08%) and threonine (2.59 mg/100 mg extract or 4.98%) were also detected in substantial amounts. Methionine (0.15 mg/100 mg extract or 0.29%) and tryptophan (0.19 mg/100 mg extract or 0.37%) were the least concentrated among the essential amino acids. Asparagine and glutamine were not detected in the SPE.

### 2.2. Characteristics of Silver Nanoparticles Biofunctionalized with Spirulina Protein Extract

#### 2.2.1. FTIR Spectra

Figure 1 presents FTIR spectra for the Spirulina protein extract (SPE) and the suspension formed from silver nanoparticles functionalized with this extract (AgNPs-SPE). These spectra highlight the complex interactions between the nanoparticles and protein components through shifts and changes in the intensities of specific bands.

Both spectra exhibit several characteristic bands. The bands at 3138 cm^−1^ and 3050 cm^−1^, specific to the SPE spectrum, correspond to the stretching vibrations of O-H and N-H bonds found in proteins. The band at 2853 cm^−1^ corresponds to the stretching vibrations of C-H bonds from alkyl groups and is common in aliphatic side chains of proteins. The bands at 1657 cm^−1^ and 1618 cm^−1^ are crucial for characterizing the secondary structure of proteins. They are specific to Amide I and Amide II, corresponding to the stretching vibrations of C=O and N-H bonds in peptide structures. The band at 1403 cm^−1^ corresponds to the symmetric stretching vibrations of the COO^−^ group in amino acids.

In the “fingerprint” region, the band at 1284 cm^−1^ is typically associated with C-N stretching vibrations in amines or Amide III. In the frequency range of 1218–1125 cm^−1^, functional groups are often associated with the stretching vibrations of C-O or C-N bonds, which appear in amide groups or etheric and alcoholic structures within proteins. Bands in the 1200–1000 cm^−1^ range are associated with C-O and C-N vibrations within the protein structure. This region provides specific details about the chemical composition of the analyzed protein extract.

The FTIR spectrum of the AgNPs-SPE mixture recorded a shift in bands in the 3200–2800 cm^−1^ region compared to the protein extract SPE. The addition of citrate-stabilized AgNPs caused a change in -OH and -NH bands, possibly due to the formation of hydrogen bonds and electrostatic interactions between proteins and nanoparticles. As a result, characteristic SPE bands, such as those at 3138 cm^−1^ and 3050 cm^−1^, were shifted or reduced in intensity in the case of AgNPs-SPE. Additionally, new bands appeared at 2920 cm^−1^ and 2936 cm^−1^ in the AgNPs-SPE spectrum, indicating interactions between Ag nanoparticles and the aliphatic side chains of the protein. This may suggest a rearrangement of C-H structures in the presence of nanoparticles.

The shift of bands from 1637 to 1639 cm^−1^ and 1532 to 1534 cm^−1^ represents changes in the Amide I and II regions, suggesting a direct interaction between AgNPs and the protein backbone. The stabilizing citrate may bind proteins through electrostatic interactions, altering C=O and N-H bonds.

The “fingerprint” region was entirely altered. For instance, the peak at 1403 cm^−1^, characteristic of the COO^−^ group in proteins, diminished in the presence of nanoparticles, suggesting interactions with citrate. The band at 1390 cm^−1^, which could be attributed to O-H interactions, reflects a modification in protein structure after interaction with citrate-stabilized nanoparticles. The band at 1235 cm^−1^, corresponding to N-H vibrations, may indicate changes in the amide bonds of the protein due to the formation of the complex with nanoparticles.

In the presence of silver nanoparticles, the bands at 1284 cm^−1^ and 1218–1125 cm^−1^ changed in intensity due to the formation of electrostatic and/or hydrogen bonds between AgNPs and the protein substrate, which altered the initial vibrations of the C-N or C-O bonds.

In the 1062–1069 cm^−1^ region, associated with the S=O group in proteins, the peak at 1069 cm^−1^ was accentuated, suggesting the binding of nanoparticles to sulfate functional groups in Spirulina proteins.

#### 2.2.2. UV-VIS

The UV-Vis spectra presented in Figure 2 display characteristic maxima for protein extract obtained from Spirulina biomass (SPE) and highlight the alteration (deformation) of the spectrum in the functional mixture AgNP-SPE. The peaks identified at 230 nm and 280 nm (Figure 2A) are characteristic of proteins derived from Spirulina biomass, with the peak at 280 nm attributed to the presence of aromatic amino acids, reflecting the presence of these structures in the tested protein extract. The deformation of the absorption spectrum, with variations appearing in the 340–400 nm range (Figure 2B), can be attributed to interactions between the silver nanoparticles and the protein components in the Spirulina extract. The UV-Vis spectrum is crucial for characterizing the specific plasmon resonance of silver nanoparticles. These small fluctuations suggest interactions between the nanoparticles and the biological matrix.

#### 2.2.3. Antioxidant Activity Stability of SPE and AgNPs-SPE

The antioxidant activity of the SPE and the AgNPs-SPE mixture was monitored over 30 days from the preparation of the mixture. It was established that the functional mixture underwent a stabilization period lasting 5 days. Initially, the mixture exhibited antioxidant values 47.7% higher than the protein matrix. On the third day, the reduction capacity of AgNPs-SPE was 82.6% above the value of SPE. By the fifth day, the antioxidant activity had decreased and was 27.3% higher. Subsequently, monitoring of antioxidant activity showed stability in the DPPH test values. The DPPH reduction capacity of AgNPs-SPE was 34.4–42% higher than the antioxidant activity of SPE (Figure 3).

### 2.3. Silver Accumulation in Animal Organs

The content of silver in animal organs following administration of AgNPs and AgNP-SPE is presented in Figure 4.

Silver was detected in all examined organs of the laboratory animals. As observed in Figure 4A, the metal from AgNP-SPE accumulated in organs in larger quantities. The highest silver content was found in brain tissue, followed by the liver, testes, spleen, kidneys, and ovaries. Similarly, silver from non-functionalized nanoparticles also accumulated significantly in brain tissue, followed by the liver, spleen, ovaries, kidneys, and testes. In the case of nanoparticles functionalized with Spirulina protein extract, the silver detected in the liver was 2.76 times higher (*p* < 0.01) compared to the silver content from non-functionalized nanoparticles in the same organ. A similar effect was observed for the amount of metal accumulated in the testes, which was 2.12 times higher (*p* < 0.05) following the administration of AgNP-SPE. The kidneys ranked second to last in silver accumulation in both groups.

Figure 4B presents the results of eliminating non-functionalized AgNPs from organs at the end of the 28-day clearance period. Minimal amounts of silver were eliminated from brain tissue. Although the difference in silver content before and after the clearance period was statistically significant, only 10.7% of the accumulated silver was eliminated from the brain. The amount of metal in the liver decreased by 51.2% and in the testes by 57.3%. Silver was eliminated from the kidneys and ovaries.

Figure 4C shows the results obtained after the clearance period following the administration of AgNP-SPE. Silver was eliminated from the liver, kidneys, and spleen. In the brain, 24% of the silver was eliminated (*p* < 0.05). In the ovaries, 24.3% (*p* < 0.05) of the accumulated metal remained, and in the testes, 12.3% (*p* < 0.05) remained.

### 2.4. Hematological and Biochemical Indicators in Animals

#### 2.4.1. Hematological Indicators

Table 2 shows the results of hematological tests conducted on animals from the four groups after 28 days of nanoparticle administration. The tests were performed immediately after the animals were sacrificed.

Table 2 indicates that, in the group of female rats administered AgNPs, hematological indicators showed a slight decrease in hemoglobin levels and erythrocyte (RBC) count (*p* < 0.05). A 33.0% increase in monocyte (MON) count (*p* < 0.05) and a 12.8% decrease in reticulocyte (RET) count (*p* < 0.05) were observed. In male rats, a 21.5% increase in platelet (PLT) count (*p* < 0.05) and a 30.5% reduction in polymorphonuclear neutrophils (PMN) (*p* < 0.05) were recorded. Both male and female rats showed an increase in eosinophil (EOS) and basophil (BAS) counts. In males, the EOS count increased by 86.52% (*p* < 0.05) and the BAS count increased 6.6-fold (*p* < 0.01), while in females, the EOS count increased by 19.9% and the BAS count by 36.3%. In the female group, the monocyte count was 34.7% lower (*p* < 0.05) compared to the control group, while in males, this indicator was 39% higher. In males, the reticulocyte count decreased by 22% (*p* < 0.05), whereas in females, it increased by 20%. In the female rats administered the AgNP-SPE, hematological indicators showed a slight reduction in hemoglobin levels and erythrocyte count (*p* < 0.05). A 44.0% increase in PMN count was observed (*p* < 0.05). An increase in BAS and RET counts was noted in both male and female rats. In males, the BAS count increased 4.84-fold (*p* < 0.05), while in females, it doubled (*p* < 0.05). The reticulocyte count showed a 20.4% increase (*p* < 0.05) in the male group and a 90% increase (*p* < 0.05) in the female group.

#### 2.4.2. Biochemical Indicators

Table 3 shows the biochemical test results for animals from the four groups after 28 days of nanoparticle administration. The tests were performed immediately after the animals were sacrificed.

An increase in blood creatinine levels was observed in the experimental animal group. In males, the increase was 34% (*p* < 0.05), while in the female group, it was 2.5 times higher (*p* < 0.05). Elevated liver function indicators were also recorded (Table 3). ALT levels in males were 36.8% higher (*p* < 0.05) and 95% higher (*p* < 0.05) in females compared to the control group. AST levels increased by 29.7% (*p* < 0.05) in males and by 2.2 times (*p* < 0.05) in females.

A rise in blood creatinine levels was also observed in the rats administered AgNPs functionalized with Spirulina protein extract. In males, creatinine levels were 67% higher (*p* < 0.01), and in females, it was 78.5% higher (*p* < 0.05). In the female group, blood glucose levels were reduced, which remained within normal limits. Additionally, elevated liver function indicators were determined. In males, ALT levels did not change. In females, ALT increased by 35.4% (*p* < 0.01). AST levels increased by 57.3% (*p* < 0.05) in males and by 2.2 times (*p* < 0.05) in females.

#### 2.4.3. Hematological and Biochemical Indices in Animals after 28 Days of AgNPs Administration and Following the 28-Day Clearance Period

Calculations were performed for the entire group, which consisted of three males and two females, to analyze the changes in hematological and biochemical indices in animals. For an appropriate comparison, values were also calculated for the experimental group of four males and three females, sacrificed 28 days after nanoparticle administration. The results of these calculations are presented in Table 4 and Table 5.

Hematological tests performed on rats treated with citrate-stabilized AgNPs confirmed the persistence of changes observed during nanoparticle treatment: decreased levels of leukocytes and polymorphonuclear cells, and increased eosinophil levels. Basophil levels returned to those of the negative control group. In animals treated with AgNPs functionalized with Spirulina protein extract, the number of polymorphonuclear cells, basophils, and reticulocytes was restored after the clearance period. A moderate increase in leukocyte count was also observed, a process that began during the administration period, leading to a 23.3% increase compared to the positive control group (Table 4).

After the clearance period, biochemical indicators evaluated in animals showed restoration of renal and hepatic parameters. In animals treated with citrate-stabilized AgNPs, clearance results indicated normal creatinine and alanine aminotransferase levels, similar to those in the negative control group. However, AST levels remained elevated among the liver enzymes. Except for AST, all other biochemical indicators returned to normal values, comparable to those in the control group (Table 5).

The clearance period normalized creatinine levels in animals treated with AgNPs functionalized with Spirulina protein extract. However, aminotransferase levels did not fully recover, with ALT remaining relatively elevated by 12.8% compared to the animals in the C (+) group. AST levels showed a continuous and significant increase of 41.7% (*p* < 0.05) compared to AST levels in animals treated for 28 days, which is 2.47 times higher than the positive control group (Table 5).

## 3. Discussion

The concept of nanoparticle functionalization is not new, with extensive research documenting the synthesis of nanoparticles using reducing substrates of plant or microbial origin, as well as their functionalization through accumulation in the structure of microorganisms during the growth cycle.

Additionally, the functionalization of nanoparticles with organic or bioactive compounds to reduce toxicity and increase their bioavailability is well known [29]. Moreover, recent studies have shown that biologically active substances are conjugated with nanoparticles to facilitate their targeted delivery to specific organs while maintaining the desired therapeutic effect [30].

Previous research achieved the biofunctionalization of AgNPs within Spirulina biomass by cultivating the cyanobacteria in a mineral medium supplemented with AgNPs [27]. This method led to the integration of nanoparticles into the biomass structures, providing insights into the bioaccumulation and potential toxicity of AgNPs when internalized by Spirulina during growth. The biomass administered to rats resulted in the accumulation of nanoparticles in different organs and their complete elimination, except for the brain [27].

Spirulina is well known for its numerous health benefits, and extracts from the biomass of this cyanobacterium have been extensively studied as nutraceuticals and in other applications. For example, organic solvent extracts from Spirulina biomass have demonstrated antibacterial, antioxidant, anti-inflammatory, and antidiabetic effects [31]. Protein extracts from Spirulina can be utilized in clinical applications to harness their immunomodulatory, antioxidant, and anti-inflammatory properties [17,32]. The protein compounds in Spirulina positively influence endothelial function and are recognized for their effectiveness in preventing and managing cardiovascular diseases [25]. The ability of Spirulina protein extract to influence nanoparticle accumulation in various tissues and enhance their elimination depends on the structural stability of the functional mixture formed. The stability of the bonds between the proteins in the protein extract and silver nanoparticles is crucial, as these bonds can impart new properties to the nanoparticles, distinct from those of synthetic AgNPs [33]. In this study, the experimental conditions used to obtain the functional mixture favored the formation of electrostatic and hydrogen bonds between the protein structures and citrate-stabilized silver nanoparticles, as evidenced by the changes observed in the UV-Vis and FTIR spectra. Additionally, the AgNPs-SPE functional mixture demonstrated a stable antioxidant activity. In a protein mixture, proteins interact with AgNPs, stabilizing them and preventing their aggregation. This stabilization reduces the likelihood of undesired redox reactions, thereby maintaining consistent values obtained in the antioxidant assay with DPPH.

The most conclusive evidence of the functional mixture’s efficacy was observed in the differential accumulation of silver in the tested organs, compared to the values obtained for simple silver nanoparticles.

In the present study, after the functionalization of silver nanoparticles with Spirulina protein extract and the characterization of their properties, these nanoparticles were administered to rats as a functional mixture with the synthesis matrix. The dosage of nanoparticles administered to the rats was 1 μg per animal, which was selected based on previous studies [27]. Higher doses of silver nanoparticles are inherently toxic, and their adverse effects have been well documented. This low dose was chosen to replicate levels similar to those used in medical applications or encountered in consumer products. The accumulation of unmodified silver nanoparticles and those in the mix with Spirulina protein extract was evaluated in various organs, including reproductive tissues. Both types of nanoparticles accumulated significantly in brain tissue, quickly crossing the blood–brain barrier. The amount of silver collected in the brain was substantial, which is highly concerning, indicating a potential risk for humans, particularly in AgNPs’ use in cosmetics and food packaging. After a 28-day clearance period, 11.4% to 24% of the accumulated silver was eliminated from the brains of the animals. The remaining silver content in the brain was approximately equal for both types of nanoparticles. Other researchers have also reported the persistence of accumulated silver in the brain even after a clearance period, highlighting its slow elimination from this organ [34]. Even with low concentrations, the maximum amount of nanoparticles was detected in brain tissue. It has been demonstrated that AgNPs administered via subcutaneous injection penetrate the rat brain and accumulate as nanoparticles [35].

Citrate-stabilized AgNPs accumulated in all organs, liver, spleen, ovaries, kidneys, and testes, with concentrations ranging between 7.9 and 11.63 ng/g. This effect is supported by other studies, which have established that regardless of the administration route—oral, intravenous, inhalation, intranasal application, or absorption through the skin—once AgNPs enter the body, they are distributed through the systemic circulation to organs such as the liver, kidneys, lungs, spleen, heart, gonads, and brain [34,36,37]. In the case of oral administration, nanoparticles cross the intestinal barrier and are absorbed from the gastrointestinal tract either as nanoparticles or in ionic form [38].

AgNPs functionalized with Spirulina protein extract exhibited specific affinity (tropism) for liver tissue, where they accumulated in amounts similar to those in the brain, three times higher than the accumulation of silver from engineered AgNPs, followed by the testes and spleen. Proteins bound to nanoparticles may interact with specific receptors on liver, kidney, and spleen cells, thereby facilitating their accumulation in these excretory organs. Regarding the route of NP administration, the oral pathway involves the capture of NPs by the reticuloendothelial system of the liver, leading to their retention for eventual elimination [14].

Due to their small size and protein functionalization, silver nanoparticles of 10 nm accompanied by Spirulina protein components could induce a tolerogenic immune response through specific interaction with liver cells. Functionalization with Spirulina protein components might enhance the specificity of these nanoparticles for immune cells and help modulate the immune response. However, potential toxicity risks associated with the small size must be considered, and safety and biodistribution studies should be conducted to ensure their safe use in therapeutic applications [39].

This study observed that the testes accumulated significant concentrations of silver from the functionalized AgNPs-SPE mixture. This suggests a possible increase in nanoparticle affinity for the functionalized mixture’s protein component. In the literature, some authors have attempted to mitigate this effect of AgNPs by biofunctionalizing them with plant extracts, but the results were contrary. Silver nanoparticles continued accumulating in the testes even after a post-administration period, thus confirming their toxicity to testicular tissue, regardless of their origin [40]. Some authors have concluded that in the brain and testes, unlike other organs, AgNPs are prone to persistence [34,41].

However, it has been demonstrated that the coating of nanoparticles can influence their tropism for the testes. For example, silver nanoparticles coated with polyethylene glycol did not exhibit affinity for the testes when administered in similar doses. On the other hand, AgNPs functionalized with live Spirulina platensis culture, orally administered to rats, significantly accumulated in the testes but were eliminated entirely over time [27]. In the case of AgNPs-SPE administration, complete elimination of silver from the testes was not observed, suggesting an increased affinity for the protein component of the functional mixture. The affinity of testicular tissue for AgNPs functionalized with Spirulina protein extract is determined by its components and can be explored for targeted drug delivery.

In the current study, it appears that the clearance period was insufficient for eliminating silver from the testes in the case of AgNPs-SPE. During the 28-day observation period, 87.7% (*p* < 0.05) of the accumulated metal was eliminated. It is possible that extending the clearance period could lead to the complete elimination of silver from testicular tissues. A similar increased affinity was observed for ovarian tissue. Once again, the silver that accumulated through the administration of citrate-stabilized AgNPs was eliminated entirely from ovarian tissue, while in the case of AgNPs-SPE, 68.7% (*p* < 0.05) was eliminated.

The silver from AgNPs-SPE was completely eliminated from the liver, spleen, and kidneys, demonstrating their stability within the protein extract structure and in the circulatory system. This allowed for more efficient transport to excretory organs, in contrast to non-functionalized AgNPs, which are less likely to be quickly recognized and captured by the reticuloendothelial system, allowing them to circulate longer in the bloodstream and be eliminated more slowly. Another hypothesis is that the protein material may be metabolized and eliminated more rapidly, reducing the potential for long-term toxicity [42].

In the case of citrate-stabilized AgNPs, elimination from the liver was not complete. This could be due to the formation of Ag ions, which are more challenging to eliminate from excretory organs [3].

Eliminating silver nanoparticles through the liver and kidneys was associated with increased levels of specific markers, indicating compromised function of these organs. In the present study, a significant increase in blood creatinine levels was observed after the administration of both AgNPs, with the highest values, and AgNP-SPE, with toxicity being more pronounced in female rats. Another study established the dependence of nanoparticle elimination from the kidneys on ovarian hormonal activity [43].

Significant increases in hepatic indicators were observed. ALT levels increased significantly in females, while in males treated with AgNPs-SPE, the values were similar to those of the control group. AST levels were elevated in males and extremely high in females. It can be noted that hepatic indicators were affected by the gender of the rats rather than by the amount of silver accumulated in the liver.

The clearance period restored ALT values in both experimental groups, but this was not the case for AST, which remained elevated. This marker remained high even in the rats treated with the AgNPs-SPE, which may indicate delayed or long-term effects.

In this study on rats treated with AgNPs, hematological indicators showed a significant increase in eosinophils and basophils, suggesting a marked inflammatory response. In females, a slight reduction in hemoglobin levels and erythrocyte count was observed, along with a significant increase in monocyte count and a decrease in reticulocytes. In the group treated with AgNPs-SPE, the changes were even more pronounced, with significant increases in the number of polymorphonuclear cells, basophils, and reticulocytes, indicating the potential effects of the treatment on the maturation and function of blood cells. After the clearance period, an increase in lymphocyte count was observed in females, which may be attributed to a persistent inflammatory process [44]. These changes highlight the potential impact of silver nanoparticles on the hematological system, requiring further evaluation to fully understand their mechanisms and implications. A similar response was observed in laboratory animals treated with AgNPs (PEG) and AgNPs functionalized with Spirulina biomass [27]. It is important to note that this immune system response is reversible, with the restoration of hematological indicators.

## 4. Materials and Methods

### 4.1. Preparation of Spirulina Protein Extract

The cyanobacterium strain *Spirulina platensis* CNMN-CB-02 biomass was used to obtain the protein extract. This strain is stored in the National Collection of Non-Pathogenic Microorganisms at the Institute of Microbiology and Biotechnology, Technical University of Moldova. Spirulina was cultivated under photoautotrophic conditions in a mineral medium containing the necessary macro- and microelements for growth and biomass accumulation in 1000 mL Erlenmeyer flasks with a suspension volume of 500 mL for 6 days. After separating the biomass from the culture medium, it was demineralized with a 1.5% ammonium acetate solution (Sigma-Aldrich Chemie GmbH, Taufkirchen, Germany) and standardized to a concentration of 100 mg/mL. The protein content in the biomass was 68–72%.

For protein extraction, the biomass was mixed with a 0.45% sodium hydroxide solution (Sigma-Aldrich Chemie GmbH, Taufkirchen, Germany) at a ratio of 1 g of biomass to 0.5 L of solution (g/L). Protein extraction was performed at 25 ± 1 °C with continuous stirring for 60 min. The supernatant was separated by centrifugation. A second extraction was performed with 200 mL of 0.45% sodium hydroxide solution for 30 min at the same temperature. The first and second extracts were combined, and the final mixture was dialyzed to a pH of 7.0–7.2. The final protein extract (SPE) was standardized to a concentration of 1% dry substance, with a protein content of 52%.

### 4.2. Functionalization of AgNPs with the Formation of the AgNPs-SPE Functional Mixture

To prepare the functional mixture of SPE and AgNPs, 10 nm silver nanoparticles stabilized in citrate, 0.02 mg/mL (Sigma-Aldrich Chemie GmbH, Taufkirchen, Germany), were used. Specification for AgNPs: 10 ± 0.2 nm particle size TEM (730786, Lot # MKCK8345). A total of 20 mL of AgNP suspension was added to 200 mL of 1% SPE solution. The mixture was stirred slowly at 200 rpm at 25 ± 1 °C for 120 min. The resulting AgNPs-SPE functional mixture contained 1.0 mg/mL SPE and 1 µg/mL AgNPs.

### 4.3. Animals and Experimental Design

The animal experiments were conducted in the Vivarium of the Laboratory of Stress Physiology, Adaptation, and General Sanocreatology at the Institute of Physiology and Sanocreatology in Moldova, in compliance with Law No. 211/2017 of the Parliament of the Republic of Moldova on the protection of animals used for experimental or scientific purposes (published in the Official Gazette No. 1–6, Article 02 on 5 January 2018), which transposes Directive 2010/63/EU of the European Parliament and of the Council of 22 September 2010 on the protection of animals used for scientific research, published in the Official Journal of the European Union L 276 on 20 October 2010. The Institute of Physiology and Sanocreatology Ethics Committee in Moldova (IREC) approved the experiments (approval IREC No. IREC/26/06.02.2023).

In the experiment, 48 Wistar albino rats (28 males and 20 females) were used, divided into four groups, each consisting of 7 males and 5 females, housed in separate cages. The animals were acclimatized under standard conditions (temperature of 22 ± 2 °C, relative humidity of 55 ± 5%, 12 h light/12 h dark photoperiod). Throughout the experiment, all animals were maintained under standard conditions with free access to water and a regular diet. The experimental groups received the following supplements: C (−)—negative control group with regular diet; C (+) positive control group with regular diet mixed with Spirulina protein extract; experimental group 1 (AgNPs)—regular diet mixed with silver nanoparticles; experimental group 2 (AgNPs-SPE)—regular diet supplemented with the mixture of silver nanoparticles functionalized with Spirulina protein extract.

The animals were fed according to these dietary regimens for 28 days in the institute’s vivarium. After this period, four males and three females from each group were sacrificed to collect blood for hematological and biochemical analyses and to harvest organs (brain, liver, spleen, kidneys, testes, and ovaries) for ICP-MS analysis. The remaining animals were maintained under optimal conditions with a regular diet for an additional 28 days (clearance period), after which they were sacrificed according to the described procedure.

The amount of AgNPs administered to the animals in experimental groups 3 and 4 was 1 µg of silver per day per animal. The positive control group received the same amount of Spirulina protein extract as the experimental group 4. The non-functionalized nanoparticle suspension and the AgNPs-SPE mixture were incorporated into whole rye flour breadcrumbs and administered to the rats as their first meal.

### 4.4. Methods for Characterizing AgNPs-SPE and Detecting Silver

#### 4.4.1. UV-Vis Spectra Recording

Samples of Spirulina protein extract and AgNPs mixture with Spirulina protein extract were prepared by dilution with distilled water. The spectra were recorded using a UV-Vis Spectrophotometer 80T (PG Instruments Ltd., Lutterworth, UK) within the wavelength range of 190–800 nm, with a spectral resolution of 1 nm.

#### 4.4.2. Fourier Transform Infrared (FTIR) Analysis

The analyzed samples were dehydrated at 40 °C. The IR spectra in the region of 4000–400 cm^−1^ were recorded at room temperature using a Perkin-Elmer FTIR spectrometer (PerkinElmer Inc., Waltham, MA, USA) equipped with an air-cooled DTGS (Deuterated Triglycine Sulfate) detector.

#### 4.4.3. ICP-MS Analysis

Before analysis, organs were dried at 70 °C until a constant weight. Next, 0.1 g of each sample were mixed with 3 mL of HNO_3_ (Sigma-Aldrich, Taufkirchen, Germany) and 1 mL of H_2_O_2_ (Sigma-Aldrich, Germany). Sample digestion was performed in an ETAS-6 analytical autoclave with resistive heating at 180 °C, pressure at 20 bar, power at 400 W, during 1.5 h. After cooling, samples were quantitatively transferred into 20 mL flasks and diluted to volume with deionized water. Silver content in samples was measured by ICP-MS using XSeries II (Thermo Scientific, Waltham, MA, USA). The uncertainty of measurements varied from 5 to 10%.

#### 4.4.4. Determination of Antioxidant Activity

The non-biological DPPH (1,1-diphenyl-2-picrylhydrazyl) radical reduction method was used. A 0.05 mM DPPH solution was prepared in 55% ethyl alcohol. The reaction mixture contained 2.9 mL DPPH solution and 0.1 mL sample. The incubation duration was 30 min. The absorbance of the samples was recorded at 517 nm. The value of the antioxidant activity was reported as % inhibition of DPPH [45].

### 4.5. Blood Hematology and Biochemistry

Hematology was evaluated using a Sysmex XT-2000i automated hematology analyzer (GMI Inc., Ramsey, MN, USA) with preset settings for analysis. Blood biochemistry was performed using a semi-automatic photometer, StarDust MC15 (DiaSys Diagnostic Systems, Holzheim, Germany).

### 4.6. Statistical Analysis

All experiments were conducted in triplicate. The results in all tables and histograms are presented as mean values ± standard deviations. Statistical differences between the values were evaluated using Student’s *t*-tests.

## 5. Conclusions

This study’s conclusions on the impact of silver nanoparticles functionalized with Spirulina protein extract (AgNPs-SPE) on rat health include significant observations related to their distribution, accumulation, and biological effects. The nanoparticles showed variable distribution across organs, including the brain and reproductive organs, demonstrating their ability to cross biological barriers such as the blood–brain barrier. Significant accumulation and persistence of silver in reproductive organs suggest potential long-term toxicity. The functionalization process influences the rates of nanoparticle accumulation and elimination. Thus, biofunctionalized silver nanoparticles accumulate in significantly higher amounts in the brain, liver, spleen, kidneys, and testes and are completely eliminated from the liver and spleen, unlike non-functionalized nanoparticles. The persistence of both types of nanoparticles in the sexual glands after a 28-day clearance period raises reasonable concerns regarding the health of the offspring of animals exposed to low levels of nanoparticles.

Both AgNPs and AgNPs-SPE affected hematological parameters, indicating possible inflammatory reactions and changes in cell maturation processes. 

The administration of AgNP-SPE induced moderate changes in hematological parameters, as evidenced by increased polymorphonuclear cells and reticulocytes in females. Additionally, the significant increase in eosinophils and basophils indicates an inflammatory response and activation of the immune system, suggesting that the body recognized the functionalized nanoparticles as foreign agents. After the 28-day clearance period, these hematological indicators returned to normal levels, indicating that the immune and inflammatory effects of AgNP-SPE are transient and reversible. Thus, eliminating the nanoparticles from the body appears to restore immune balance.

The administration of AgNP-SPE also induced transient biochemical changes, particularly related to liver and kidney function. After the clearance period, most indicators returned to normal levels, except for AST, suggesting potential residual effects on the liver.

The impact of AgNPs on the liver and kidney and their persistence in these vital organs requires further evaluation to determine the safety and appropriate dosing of AgNPs-SPE. Specifically, studies should investigate the long-term effects on hepatic and renal tissue, the exact localization of AgNPs within reproductive tissues, their impact on reproductive function, and the duration of their persistence in these organs.

## Figures and Tables

**Figure 1 pharmaceuticals-17-01247-f001:**
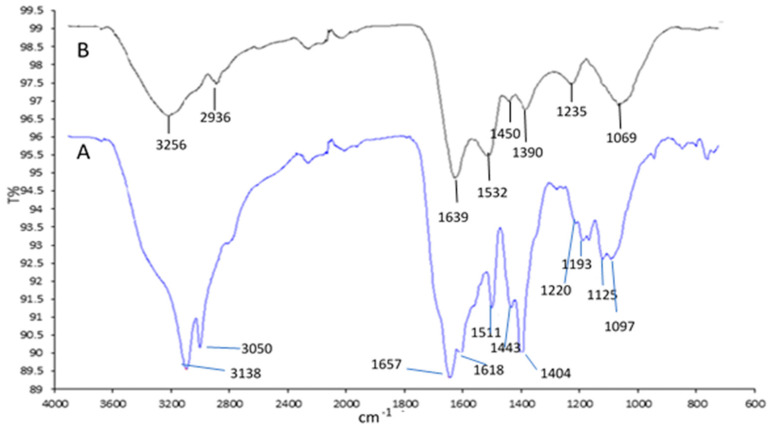
FTIR spectra of (A) Spirulina protein extract (SPE) and (B) the functional mixture AgNPs-SPE.

**Figure 2 pharmaceuticals-17-01247-f002:**
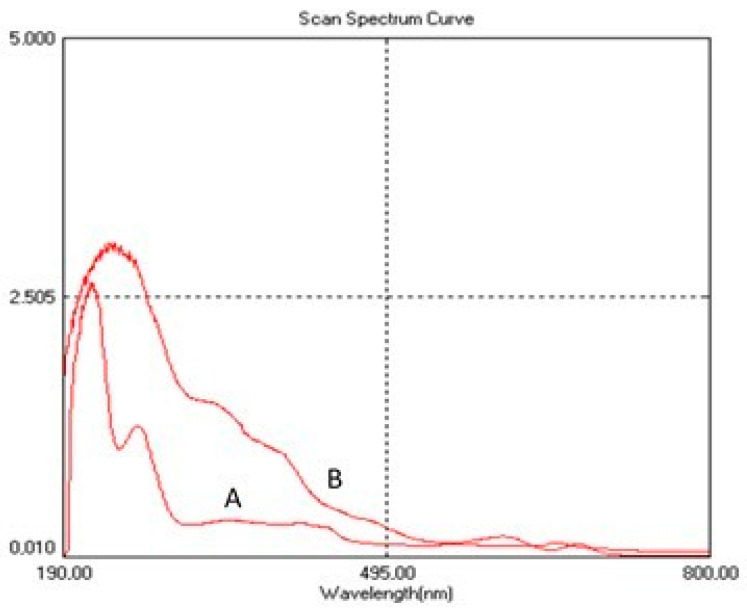
UV-Vis absorption spectra of (A) Spirulina protein extract (SPE) and (B) functional mixture AgNPs-SPE.

**Figure 3 pharmaceuticals-17-01247-f003:**
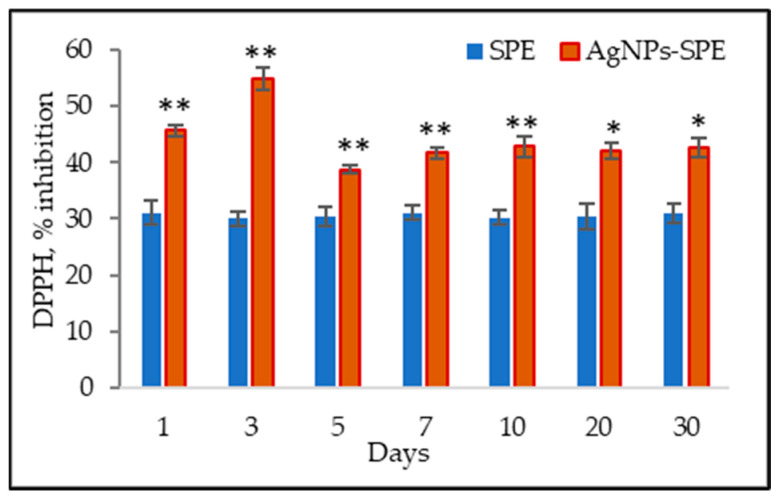
Antioxidant activity (DPPH, % inhibition) of Spirulina protein extract (SPE) and functional mixture (AgNPs-SPE) monitored over 30 days. * *p* ˂ 0.05 and ** *p* ˂ 0.01 indicate a significant difference between adjacent groups.

**Figure 4 pharmaceuticals-17-01247-f004:**
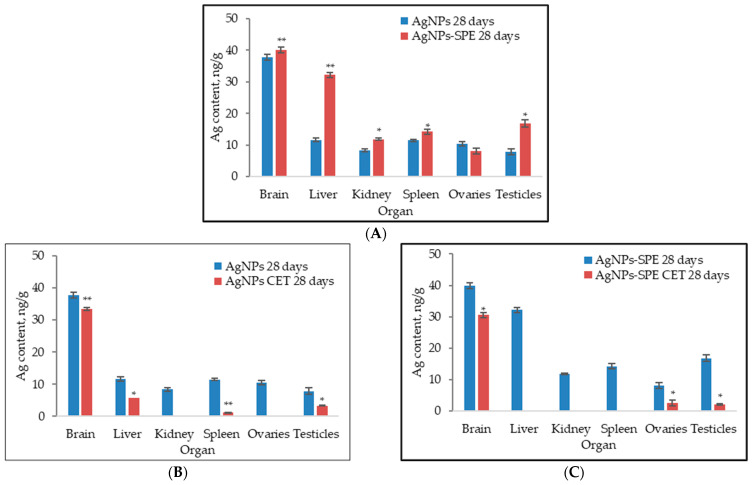
The content of silver in rats’ organs: (**A**) animals administered AgNPs and AgNPs-SPE for 28 days, measured immediately after the end of the experiment; (**B**) animals administered AgNPs for 28 days with measurements taken after a clearance expiration time (CET, 28 days); (**C**) animals administered AgNPs-SPE for 28 days with measurements taken after a clearance expiration time (CET, 28 days). * *p* ˂ 0.05 and ** *p* ˂ 0.01 indicate a significant difference between adjacent groups.

**Table 1 pharmaceuticals-17-01247-t001:** Amino acids composition of the Spirulina protein extract (SPE).

Amino Acid	Protein Extract from Spirulina (SPE)	
	mg/100 mg Extract	% Total Extract
Alanine	4.28	8.23
Arginine	5.58	10.73
Asparagine	0.00	0.00
Aspartic acid	5.02	9.66
Cysteine	1.14	2.19
Glutamic acid	9.04	17.39
Glutamine	0.00	0.00
Glycine	2.88	5.54
Histidine *	0.81	1.56
Isoleucine *	0.08	0.15
Leucine *	5.12	9.85
Lysine *	3.16	6.08
Methionine *	0.15	0.29
Phenylalanine *	2.27	4.37
Proline	2.03	3.90
Threonine *	2.59	4.98
Tryptophan *	0.19	0.37
Serine	3.06	5.89
Tyrosine	2.68	5.15
Valine *	1.91	3.67
Total	51.99	100.00

* Essential amino acids.

**Table 2 pharmaceuticals-17-01247-t002:** Hematological indices in rats.

**Experimental Group**	**C (−)**		**AgNPs**	
**Animal Sex**	**Male**	**Female**	**Male**	**Female**
HB g/L	153.25 ± 5.32	160.67 ± 4.04	155.0 ± 1.15	152.30 ± 2.31 *
RBC, 10^12^/L	8.73 ± 0.34	8.83 ± 0.28	8.58 ± 0.21	8.14 ± 0.09 *
WBC, 10^9^/L	18.07 ± 2.41	15.76 ± 2.51	16.75 ± 3.12	13.56 ± 0.84
PLT, 10^9^/L	505.0 ± 101.22	626.0 ± 232.0	613.50 ± 32.21 *	633.0 ± 43.30
PMN, %	31.79 ± 6.80	24.33 ± 4.15	22.08 ± 6.17 *	24.0 ± 0.17
LY, %	56.30 ± 8.12	58.87 ± 3.35	60.68 ± 3.99	61.2 ± 5.02
MON, %	6.03 ± 1.84	9.50 ± 0.46	8.38 ± 0.61 *	6.20 ± 1.25 *
EOS, %	4.45 ± 0.30	6.73 ± 1.50	8.3 ± 2.55 *	8.07 ± 1.95 *
BAS, %	0.58 ± 0.33	0.47 ± 0.38	3.83 ± 0.60 **	0.64 ± 0.23 *
RET, %	3.35 ± 0.37	3.35 ± 1.81	2.92 ± 0.13 *	4.03 ± 0.44
**Experimental Group**	**C (+)**		**AgNPs-SPE**	
**Animal Sex**	**Male**	**Female**	**Male**	**Female**
HB g/L	151.25 ± 5.50	151.33 ± 4.51	152.75 ± 3.50	138.0 ± 10.39 *
RBC, 10^12^/L	8.11 ± 0.43	8.19 ± 0.41	8.41 ± 0.13	7.87 ± 0.31 *
WBC, 10^9^/L	12.59 ± 3.69	12.92 ± 4.08	14.24 ± 3.19	12.02 ± 2.23
PLT, 10^9^/L	523.25 ± 116.25	672.0 ± 77.09	490.75 ± 125.65	643.33 ± 198.94
PMN, %	25.08 ± 9.41	24.77 ± 1.46	26.63 ± 2.55	35.67 ± 9.58 *
LY, %	56.40 ± 11.97	62.47 ± 2.63	53.83 ± 1.99	52.17 ± 5.83
MON, %	6.70 ± 3.11	8.23 ± 1.75	7.62 ± 1.60	8.70 ± 1.73
EOS, %	5.56 ± 0.82	7.23 ± 2.29	5.38 ± 0.37	5.10 ± 0.52
BAS, %	0.68 ± 0.10	0.30 ± 0.10	3.30 ± 1.41 *	0.67 ± 0.29 *
RET, %	3.43 ± 0.45	3.97 ± 1.19	4.13 ± 0.25 *	7.54 ± 0.76 *

C (−)—negative control; C (+)—positive control (Spirulina protein extract); AgNPs—experimental group administered with silver nanoparticles; AgNPs-SPE—experimental group administered with silver nanoparticles functionalized with Spirulina protein extract; HB—hemoglobin; RBC—erythrocytes; WBC—leukocytes; PLT—platelets; PMN—polymorphomultinuclear neutrophil granulocytes; LY—lymphocytes; EOS—eosinophils; BAS—basophils; MON—monocytes; RET—reticulocyte; *n* = 4 for male; *n* = 3 for female; * *p* < 0.05; ** *p* < 0.01.

**Table 3 pharmaceuticals-17-01247-t003:** Biochemical indices in rats.

**Experiment Group**	**C (−)**		**AgNPs**	
**Animal Sex**	**Male**	**Female**	**Male**	**Female**
Prot, g/L	58.55 ± 7.09	59.53 ± 4.51	60.75 ± 3.72	57.43 ± 3.27
Glu, mmol/L	4.93 ± 0.35	6.22 ± 0.40	5.26 ± 0.64	5.76 ± 0.75
CREA, µM/L	92.76 ± 20.03	84.97 ± 4.04	161.40 ± 19.13 *	170.30 ± 16.8 **
Urea mg/dL	29.35 ± 8.76	28.62 ± 6.81	28.91 ± 4.69	26.57 ± 4.41
ALT, U/L	173.6 ± 9.08	103.45 ± 8.60	237.45 ± 25.63 *	201.83 ± 47.89 *
AST, U/L	3.57 ± 0.86	3.57 ± 0.12	4.63 ± 0.13 *	7.23 ± 1.79 *
**Experiment Group**	**C (+)**		**AgNPs-SPE**	
**Animal Sex**	**Male**	**Female**	**Male**	**Female**
Prot, g/L	66.13 ± 2.29	61.57 ± 9.74	60.73 ± 3.55	57.33 ± 9.67
Glu, mmol/L	6.33 ± 0.51	5.26 ± 0.91	5.86 ± 0.76	3.76 ± 1.22 *
CREA, µM/L	105.60 ± 4.16	94.60 ± 4.30	176.4 ± 4.16 **	168.90 ± 4.21 **
Urea mg/dL	24.75 ± 6.95	26.57 ± 5.63	27.60 ± 4.61	27.16 ± 5.33
ALT, U/L	197.45 ± 39.07	168.07 ± 67.32	178,68 ± 12.23	227.50 ± 1.66 **
AST, U/L	3.07 ± 2.22	2.53 ± 1.79	4.83 ± 5.24 *	5.13 ± 1.01 *

C (−)—negative control; C (+)—positive control (Spirulina protein extract); AgNPs—experimental group administered silver nanoparticles; AgNPs-SPE—experimental group administered silver nanoparticles functionalized with Spirulina protein extract; Prot—total protein; Glu—glucose; CREA—creatinine; ALT—alanine aminotransferase; AST—aspartate aminotransferase; *n* = 4 for male; *n* = 3 for female; * *p* < 0.05; ** *p* < 0.01.

**Table 4 pharmaceuticals-17-01247-t004:** Hematological indices in rats after 28 days of AgNPs and AgNPs-SPE administration and a 28-day clearance period.

Indices	C (−)	AgNPs	AgNPs CET	C (+)	AgNPs-SPE	AgNPs-SPE CET
HB g/L	156.43 ± 5.94	153.86 ± 2.11	154.00 ± 1.87	145.57 ± 7.14	146.43 ± 10.2	150.0 ± 1.41
RBC, 10^12^/L	8.77 ± 0.30	8.39 ± 0.28	8.41 ± 0.29	8.15 ± 0.38	8.18 ± 0.35	8.05 ± 0.50
WBC, 10^9^/L	17.08 ± 2.55	15.38 ± 2.83 *	14.84 ± 2.68	12.73 ± 3.52	13.29 ± 2.86	15.70 ± 2.08
PLT, 10^9^/L	556.8 ± 165.6	623.5 ± 37.7	621.4 ± 35.5	587.0 ± 122.7	556.1 ± 163.0	472.0 ± 159.8
PMN, %	28.59 ± 6.69	23.0 ± 3.18 *	23.66 ± 4.84	22.37 ± 7.51	30.5 ± 7.56 *	23.36 ± 4.20 *
LY, %	57.40 ± 6.21	60.9 ± 4.06	59.80 ± 4.40	62.00 ± 9.19	53.11 ± 3.76	58.74 ± 4.82
MON, %	7.56 ± 2.33	7.44 ± 1.42	8.04 ± 1.61	7.36 ± 2.55	8.08 ± 1.62	8.01 ± 2.02
EOS, %	5.43 ± 1.61	8.20 ± 2.23 *	7.85 ± 2.42	6.33 ± 1.67	5.24 ± 0.44	7.07 ± 1.23
BAS, %	0.53 ± 0.33	2.23 ± 0.42 *	0.64 ± 0.42	0.51 ± 0.22	2.17 ± 1.73 *	0.8 ± 0.77
RET, %	3.35 ± 1.08	3.39 ± 0.65	3.33 ± 0.60	3.66 ± 0.81	5.84 ± 0.50 *	4.26 ± 0.17

C (−)—negative control; C (+)—positive control (Spirulina protein extract); AgNPs—experimental group administered with silver nanoparticles; AgNPs-SPE—experimental group administered with silver nanoparticles functionalized with Spirulina protein extract; *n* = 7 for C (−), C (+), AgNPs, and AgNPs-SPE; *n* = 5 for AgNPs CET, and AgNPs-SPE CET; * *p* < 0.05 for the differences between the groups AgNPs/C (−), AgNPs CET/AgNPs, AgNPs-SPE/C (+), and AgNPs-SPE CET/AgNPs-SPE.

**Table 5 pharmaceuticals-17-01247-t005:** Biochemical indices in rats after 28 days of AgNPs and AgNPs-SPE administration and a 28-day clearance period.

Indices	C (−)	AgNPs	AgNPs CET	C (+)	AgNPs-SPE	AgNPs-SPE CET
Prot, g/L	58.97 ± 5.68	59.34 ± 3.70	68.87 ± 3.50	64.17 ± 6.47	59.27 ± 6.38	62.46 ± 4.04
Glu, mmol/L	5.49 ± 0.77	5.47 ± 0.68	5.49 ± 0.41	5.87 ± 0.86	4.96 ± 1.34	5.65 ± 0.47
CREA, µM/L	86.43 ± 14.9	165.2 ± 17.10 **	69.33 ± 3.71 **	100.93 ± 7.77	172.77 ± 5.92 *	68.04 ± 3.92 *
Urea, mg/dL	29.03 ± 5.69	27.01 ± 4.36	31.17 ± 0.36	25.53 ± 5.97	27.41 ± 4.49	33.72 ± 0.36
ALT, U/L	162.4 ± 39.6	222.0 ± 45.0 **	188.2 ± 13.5 *	184.9 ± 50.2	199.6 ± 27.0 *	208.6 ± 10.6
AST, U/L	3.57 ± 2.31	5.74 ± 1.74 *	5.43 ± 0.80	2.84 ± 1.90	4.96 ± 2.56	7.03 ± 0.68 *

C (−)—negative control; C (+)—positive control (Spirulina protein extract); AgNPs—experimental group administered with silver nanoparticles; AgNPs-SPE—experimental group administered with silver nanoparticles functionalized with Spirulina protein extract; *n* = 7 for C (−), C (+), AgNPs, and AgNPs-SPE; *n* = 5 for AgNPs CET and AgNPs-SPE CET; * *p* < 0.05 and ** *p* < 0.01 for the differences between the groups AgNPs/C (−), AgNPs CET/AgNPs, AgNPs-SPE/C (+), and AgNPs-SPE CET/AgNPs-SPE.

## Data Availability

Data are contained within the article.

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
