# Peer review of "The Impact of Silver Nanoparticles Functionalized with Spirulina Protein Extract on Rats"

_pharmaceuticals, 2024, doi:10.3390/ph17091247_

Round 1
Reviewer 1 Report
Comments and Suggestions for Authors
Present article studies the influence of silver nanoparticles functionalized with Spirulina protein extract (AgNPs-SPE) on rats through the evaluation of the systemic distribution, organ accumulation, and potential toxicity associated.
The introduction provides enough background to understand the problematic and how authors wants to solve it, and even in the discussion they add more information of interest about it. The methodology is well described and the experiments are appropiate to study the aspects mentioned about the bioaccumulation. There is enough information in the methodology to reproduce the experiments. The results syupport the conclusions exposed about the prevalence of the accumulation of the silver nanoparticles in reproductive organs and their different bioadsorption influenced by the functionalization.
I consider this articla suitable for publication in present form.
Author Response
The introduction provides enough background to understand the problematic and how authors wants to solve it, and even in the discussion they add more information of interest about it. The methodology is well described and the experiments are appropiate to study the aspects mentioned about the bioaccumulation. There is enough information in the methodology to reproduce the experiments. The results syupport the conclusions exposed about the prevalence of the accumulation of the silver nanoparticles in reproductive organs and their different bioadsorption influenced by the functionalization.
I consider this articla suitable for publication in present form.
Answer: Thank you for your kind appreciation of our work. We are grateful for your positive feedback and glad our study met your expectations. Your supportive comments are highly motivating and encourage us to continue our research with the same dedication.
Reviewer 2 Report
Comments and Suggestions for Authors
The authors adequately present the research work entitled "The Impact of Silver Nanoparticles Functionalized with Spir-ulina Protein Extract on Rat" the article is well described in general, with good scientific rigor. I have the following observations
1.- Improve the quality of the images specifically IR (wave number) and UV-Vis
2.- Rewrite 4.2 the description for the preparation of silver nanoparticles with spirulina is not clear, the nanoparticles were acquired and then slowly mixed at 2000 rpm???
3.- In 4.3.3 icp-ms analysis for the digestion of the samples they use nitric acid and hydrogen peroxide to determine the concentration of silver and for gold they use nitric acid and hydrochloric acid with gold nanoparticles. Explain why
4.- Review the digestion parameters
5.- A large number of tables are placed where the results of the hematological indices and biochemical indices are shown, these are not considered in the conclusions, I consider it necessary to demonstrate the importance of determining these indices and describe it in the conclusions
Author Response
Thank you for thoroughly reviewing our work and pointing out the errors in the presentation. We greatly appreciate your observations, which have improved the quality of the paper and resulted in a more comprehensive and nuanced conclusion. The conclusions now already reflect the complete outcome of the study. We thank you once again!
- Improve the quality of the images specifically IR (wave number) and UV-Vis
Answer: We are very sorry, but unfortunately, we cannot make further modifications. However, the IR spectra (original images) are uploaded in the supplementary material.
2. Rewrite 4.2 the description for the preparation of silver nanoparticles with spirulina is not clear, the nanoparticles were acquired and then slowly mixed at 2000 rpm???
Corrected: “the nanoparticles were slowly mixed at 2000 rpm”.
3. In 4.3.3 icp-ms analysis for the digestion of the samples they use nitric acid and hydrogen peroxide to determine the concentration of silver and for gold they use nitric acid and hydrochloric acid with gold nanoparticles. Explain why.
Answer: Use of different procedures of samples digestion is explained by the fact that gold nanoparticles demand specific mixtures, as in case of our study aqua regia for their decomposition. Silver nanoparticles do not require special conditions for digestion; therefore, the standard procedure was used. Since manuscript describes effect of silver nanoparticles on rats information about gold nanoparticles was removed from the manuscript.
4. Review the digestion parameters
Answer: Information was revised
5. A large number of tables are placed where the results of the hematological indices and biochemical indices are shown, these are not considered in the conclusions, I consider it necessary to demonstrate the importance of determining these indices and describe it in the conclusions
Answer: The conclusions have been reformulated. Information about all the tests performed and analyzed has been included. Thank you very much!

Reviewer 3 Report
Comments and Suggestions for Authors
In my review, I have several comments and suggestions that should be addressed before the manuscript can be considered:
Major comment
· Before all, give justification if any similarly between this paper and the paper that you have published in Nanomaterials (Accumulation and Effect of Silver Nanoparticles Functionalized with Spirulina platensis on Rats) https://doi.org/10.3390/nano11112992 . what are the different between these two studies? Is their any of the data have been repeated in both paper ?
Minor comments
1. In abstract, line 17, the verb using should be written before (ICP-MS analysis).
2. The objective mentions the "potential toxicity" of the biofunctionalized nanoparticles, but it could benefit from more specificity. Consider clarifying whether the study aims to compare the toxicity of functionalized nanoparticles to non-functionalized ones or simply to assess the toxicity in general.
3. Elimination" in Line 35: The statement that AgNPs are "eliminated from the body without the risk of accumulation" seems to contradict findings in your study that AgNPs accumulate in organs like the brain and testicles. Consider clarifying this point, especially since the persistence of nanoparticles in certain organs is a key part of your study.
4. Method; Did you evaluate its stability over time (e.g., through visual inspection, size distribution measurements, or zeta potential analysis)? This is important to ensure the functional mixture remains stable for subsequent experiments.
5. The daily administration of 1 μg of silver per animal is well specified, but it would be helpful to explain why this dosage was selected. Was this based on previous studies, or was it chosen based on preliminary research?
6. Line 497, containing gold nanoparticles. Their no any mention for gold in the manuscript, what did you mean by gold?
7. Methods Section: The sentence describing ICP-MS analysis to assess organ distribution could be clearer. A brief mention of how the organs were sampled and analyzed (e.g., tissue extraction and preparation for ICP-MS) might help readers unfamiliar with the technique.
8. Figure 1 legend, mention (A) and (B) in the legend.
9. You mention the appearance of maxima at 280 nm and 390 nm but do not explain in detail how these new peaks are related to the presence of silver nanoparticles (AgNP). It would strengthen your argument to reference how these peaks are characteristic of AgNP interaction with proteins or other similar studies.
10. Table 2. Biochemical Indices in Rats, the parameters included are for hematological indices, is this mistake, correct it .
11. Conclusion; further evaluation: When recommending further evaluation, specifying the type of studies (e.g., long-term toxicity studies, reproductive health assessments, or biochemical pathway investigations) would make the conclusion more actionable and precise.
Author Response
Thank you very much for your thorough review of our work. Your careful attention identified errors in the presentation of our results, which we could then eliminate. Your feedback has significantly improved the quality of the manuscript, and we are sincerely grateful for your contribution.
Major comment. Before all, give justification if any similarly between this paper and the paper that you have published in Nanomaterials (Accumulation and Effect of Silver Nanoparticles Functionalized with Spirulina platensis on Rats) https://doi.org/10.3390/nano11112992 . what are the different between these two studies? Is their any of the data have been repeated in both paper ?
Answer: While both papers investigate silver nanoparticles functionalized with Spirulina platensis and their effects on rats, they employ different methodologies with no repeated data. In the current study, AgNPs were functionalized by mixing Spirulina protein extract with 10 nm AgNPs stabilized in citrate, focusing on the impact of this mixture on biological systems. In contrast, the paper published in Nanomaterials explored biofunctionalizing AgNPs within the Spirulina biomass by cultivating the organism in a mineral medium supplemented with AgNPs. This method provided insights into the bioaccumulation and potential toxicity of AgNPs internalized by Spirulina during growth. Consequently, the results differ, with AgNP-SPE showing an increased affinity for certain tissues. A detailed description of the nanoparticle functionalization method applied in the previous study has been added to the Discussion section.
1. In abstract, line 17, the verb using should be written before (ICP-MS analysis).
Corrected
2. The objective mentions the "potential toxicity" of the biofunctionalized nanoparticles, but it could benefit from more specificity. Consider clarifying whether the study aims to compare the toxicity of functionalized nanoparticles to non-functionalized ones or simply to assess the toxicity in general.
Answer: The objective was modified in the Abstract and at the end of the Introduction section. Indeed, our aim was not to identify the toxic effects of the nanoparticles, and the selected dose supports this argument.
3. Elimination" in Line 35: The statement that AgNPs are "eliminated from the body without the risk of accumulation" seems to contradict findings in your study that AgNPs accumulate in organs like the brain and testicles. Consider clarifying this point, especially since the persistence of nanoparticles in certain organs is a key part of your study.
Answer: The sentence was removed because, indeed, it raises several unnecessary questions.
4. Method; Did you evaluate its stability over time (e.g., through visual inspection, size distribution measurements, or zeta potential analysis)? This is important to ensure the functional mixture remains stable for subsequent experiments.
Answer: The stability of the functional mixture was verified using the antioxidant test. In a protein mixture, the proteins interact with AgNPs, stabilizing and preventing their aggregation. This stabilization reduces the likelihood of undesirable redox reactions, thereby maintaining consistent values in the antioxidant assay with DPPH. DPPH is a non-biological radical that can be reduced through direct or indirect proton transfer or with another radical. The result's names were modified for clarity.
5. The daily administration of 1 μg of silver per animal is well specified, but it would be helpful to explain why this dosage was selected. Was this based on previous studies, or was it chosen based on preliminary research?
Answer: The daily dosage of 1 μg of silver per animal was selected based on previous studies. Higher doses of silver nanoparticles are inherently toxic, and their adverse effects have been extensively analyzed. By choosing a low dose, we aimed to replicate levels similar to those administered in medical applications or those that might be encountered through consumer products. The justification for the selected dose was included in the Discussion section.
6. Line 497, containing gold nanoparticles. Their no any mention for gold in the manuscript, what did you mean by gold?
Corrected: The mention of gold was removed.
7. Methods Section: The sentence describing ICP-MS analysis to assess organ distribution could be clearer. A brief mention of how the organs were sampled and analyzed (e.g., tissue extraction and preparation for ICP-MS) might help readers unfamiliar with the technique.
Answer: More details about samples preparation and analysis were added to the manuscript.
8. Figure 1 legend, mention (A) and (B) in the legend.
Corrected: The figure name was rewritten.
9. You mention the appearance of maxima at 280 nm and 390 nm but do not explain in detail how these new peaks are related to the presence of silver nanoparticles (AgNP). It would strengthen your argument to reference how these peaks are characteristic of AgNP interaction with proteins or other similar studies
Answer: The description of the spectra has been modified.
10. Table 2. Biochemical Indices in Rats, the parameters included are for hematological indices, is this mistake, correct it
Corrected: In the manuscript, the table has been replaced. Thank you very much!
11. Conclusion; further evaluation: When recommending further evaluation, specifying the type of studies (e.g., long-term toxicity studies, reproductive health assessments, or biochemical pathway investigations) would make the conclusion more actionable and precise.
Answer: The conclusions have been reformulated.

Round 2
Reviewer 3 Report
Comments and Suggestions for Authors
the authors address all my points